# Neural Network Quine

**Oscar Chang & Hod Lipson**
Department of Computer Science
Columbia University
New York, NY 10027, USA
{oscar.chang,hod.lipson}@columbia.edu

## Abstract

We describe a method for building a self-replicating neural network. We also test it on an image classification network for MNIST.

## 1 Self-Replication

John von Neumann pioneered the concept of an artificial self-replicating machine before the discovery of DNA. He described a set of initial states and transformation rules for a cellular automaton that upon running for a fixed number of steps produces copies of the initial cell states (Von Neumann & Burks, 1966). Hofstadter (1980) later coined the term 'quine' in *Gödel, Escher, Bach: an Eternal Golden Braid* after the philosopher Willard Van Orman Quine, to describe self-replicating expressions such as: 'is a sentence fragment' is a sentence fragment.

In the context of programming language theory, quines are computer programs that print their own source code. A trivial example of a quine is the empty string, which in most languages, the compiler transforms into the empty string. The following code snippet is an example of a non-trivial Python quine written in two lines.

```
s = 's = %r\nprint(s%%s)'
print(s%s)
```

There are several motivations for studying self-replicating neural networks.

- Neural networks are capable of learning powerful representations across many different domains of data (Bengio et al., 2013). But can a neural network learn a good representation of itself? Self-replication involves some level of self-awareness, and is a small step towards developing introspective capabilities in neural networks.

- Biological life began with the first self-replicator (Marshall, 2011), and natural selection kicked in to favor organisms that are better at replication, resulting in a self-improving mechanism. Analogously, self-replicating neural networks can be the precursor to self-improving neural networks.

- In a HyperNetwork (Ha et al., 2017), a small recurrent neural network is used to generate the weights for a larger one, which can be viewed as a meta-controller enforcing a soft weight-sharing constraint between layers of a recurrent neural network. Similarly, we can view self-replication as a mechanism that enforces a soft weight-sharing constraint between a network and past versions of itself, which is helpful for lifelong learning (Silver et al., 2013).

- Learning how to enhance or diminish the ability for AI programs to self-replicate is useful for computer security. For example, we might want an AI to be able to execute its source code without being able to read or reverse-engineer it, either through its own volition or interaction with an adversary.

In this extended abstract, we consider the problem of building a neural network quine, and propose a method to solve it. We also show that it is possible for a quine to perform other useful functions in addition to self-replication. Specifically, we demonstrate as a proof of concept a quine that can classify MNIST images (LeCun & Cortes, 1998). Finally, we discuss present challenges and possibilities for future work.

## 2 NEURAL NETWORK QUINE

### 2.1 PROBLEM WITH DIRECT REFERENCE

A neural network is parametrized by a set of parameters $\Theta$, and our goal is to build a network that outputs $\Theta$ itself. This is difficult to do directly. Suppose the last layer of a feed-forward network has $A$ inputs and $B$ outputs. Already, the size of the weight matrix in a linear transformation is the product $AB$ which is greater than $B$ for any $A > 1$.

We also looked at open-source implementations of two popular generative models for images, DC-GAN (Radford et al., 2016) and DRAW (Gregor et al., 2015). They use 12 million and 1 million parameters respectively to generate MNIST images with 784 pixels.

In general, the set of parameters $\Theta$ is a lot larger than the size of the output. To circumvent this, we need an indirect way of referring to $\Theta$.

### 2.2 INDIRECT REFERENCE

HyperNEAT (Stanley et al., 2009) is a neuro-evolution method that describes a neural network by identifying every topological connection with a coordinate and a weight. We pursue the same strategy in building a quine. Instead of having the quine output its weights directly, we shall set it up so that it inputs a coordinate (in a one-hot encoding) and outputs the weight at that coordinate.

This overcomes the problem of $\Theta$ being larger than the output, since we are only outputting a scalar $\Theta_c$ for each coordinate $c$.

### 2.3 SELF-REPLICATING LOSS

We define the self-replicating loss to be the sum of the squared difference between the actual weight and its predicted value.

$$L_{SR} = \sum_{c \in C} \left\| f_\Theta(c) - \Theta_c \right\|_2^2 \tag{1}$$

We propose to build a neural network quine by training it using standard stochastic gradient descent on the self-replicating loss, where we take $C$ to be a mini-batch of coordinates during training, and the set of all coordinates during testing.

## 3 EXPERIMENTS

### 3.1 VANILLA QUINE

We train a feed-forward network (vQuine) with 7 hidden layers, each mapping from 50 inputs to 50 outputs. The loss function is a moving target, since $\Theta_c$ changes after each gradient update. Nonetheless, we can achieve a loss of 93.18 (average weight prediction error of 0.07) after 100 epochs of training (Fig. 1), noting that the loss prior to any training is 6171.60 (average weight prediction error of 0.59). An exact quine should have close to zero self-replicating loss, where any loss above zero ought to be an artifact of numerical imprecision.

We find two main issues with quine training. Firstly, it is sensitive to the choice of weight initialization and activation function, and some choices may result in an exploding loss. This is a commonly encountered problem in deep learning, and does not seem specific to quine training. Secondly, an average weight prediction error of 0.07 still seems pretty significant. We speculate that other input encodings and non-gradient based training methods will be helpful in mitigating these issues.

## 3.2 MNIST Quine

It is possible to jointly optimize an existing loss function with the self-replicating loss so that an existing neural network gains the ability to self-replicate in addition to the task it was originally meant to perform.

$$L_{MNIST+SR} = L_{SR} + \lambda L_{MNIST} \qquad (2)$$

We use the same feed-forward network from before, but modify its input and output layer such that the network inputs an MNIST image and a coordinate, and outputs a classification label for the image and the weight for the coordinate. The modified network (mQuine) is trained on the loss $L_{MNIST+SR}$ in Eqn. 2 with $L_{MNIST}$ being the cross-entropy loss for the classification of the input MNIST image.

As $L_{MNIST}$ drops in the course of training, we notice that $L_{SR}$ goes up slightly (Fig. 4). This suggests that it is more difficult for a network that has increased its specialization at a particular task to self-replicate.

We set up the same feed-forward network to perform just the task of MNIST classification, and use that as a baseline with which to compare the test accuracy of mQuine (Fig. 5). It is clear that mQuine should perform worse than the baseline, because it is accomplishing the additional task of self-replication whilst using the same network. We observe that mQuine achieved 87.42% test accuracy after 900 epochs of training compared to 94.44% in the baseline, and took a significantly longer time to converge.

It is promising that the neural network in our setup is able to handle the joint optimization problem, even though self-replication occupied a significant portion of its capacity.

## 4 FUTURE WORK

There are multiple directions for future work.

- Use non-gradient based methods (e.g. an evolutionary algorithm) to optimize the self-replicating loss.
- Determine an acceptable threshold for the average weight prediction error. If using the predicted weights in place of the actual weights does not increase the self-replicating loss and repeating this process results in a fixed point, then the prediction error should not be considered significant.
- Calculate the self-replicating quotient of a neural network quine using the methodology laid out in Adams & Lipson (2009). For example, Zykov et al. (2005) estimate the self-replicating quotient of Penrose Tiling (Penrose, 1959) to be below $log 2$ and that of animals to be at least $10^{20}$.
- Investigate if the self-replicating loss might be useful for regularization.
- Explore other encoding methods and architectures. For example, we can divide the million parameters in DRAW into partitions of size 784, and output 784 weight predictions at once.
- Build Ouroboros neural networks, for instance, a pair of co-referring networks.

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

## A  FIGURES

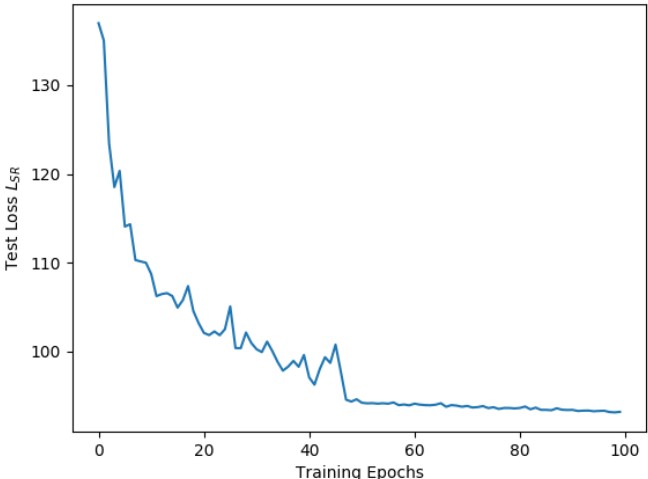

Figure 1: Plot of vQuine's test loss over 100 training epochs.

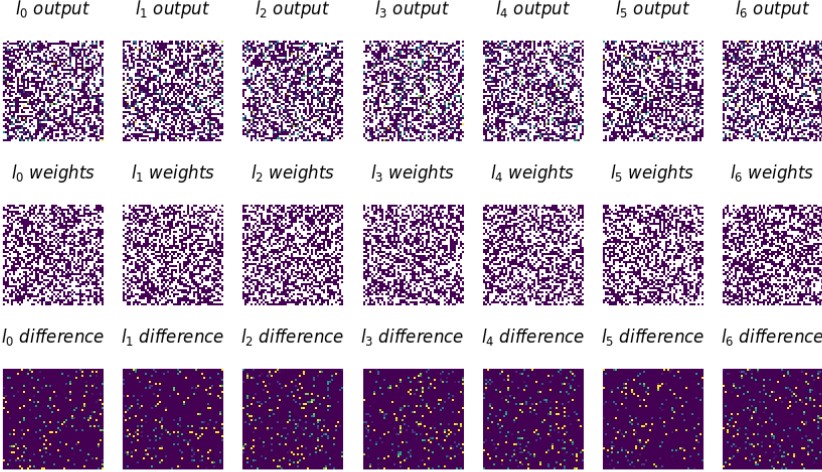

Figure 2: A log-normalized illustration of the actual weights, outputs, and the squared difference between them for the 7 hidden layers of vQuine, prior to any training.

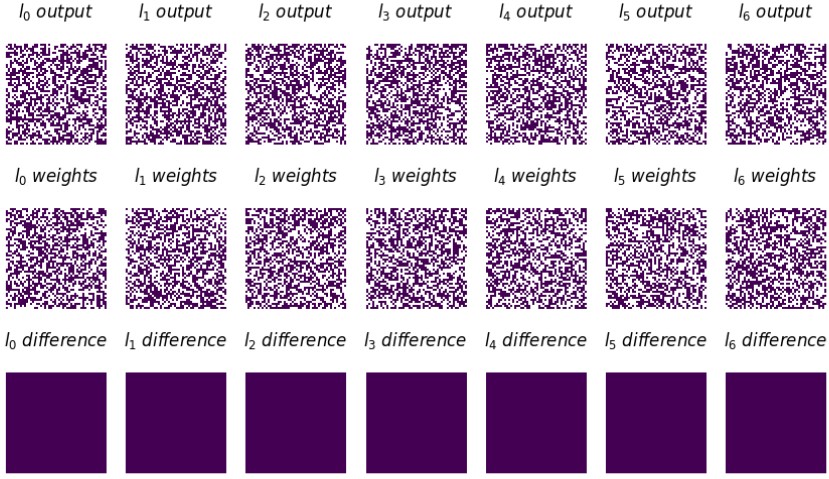

Figure 3: A log-normalized illustration of the actual weights, outputs, and the squared difference between them for the 7 hidden layers of vQuine, after 100 epochs of training.

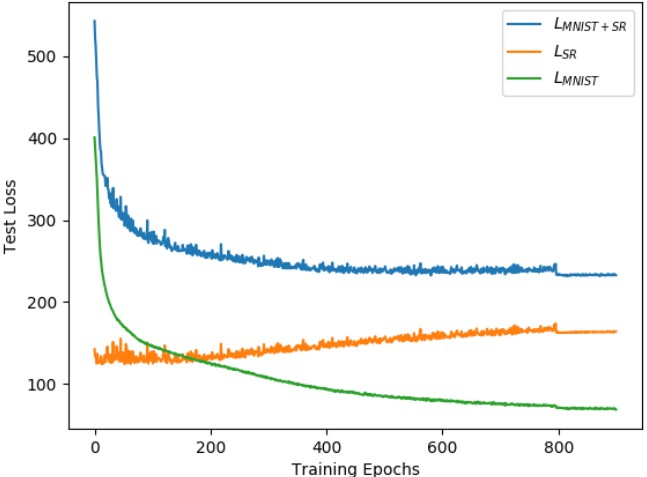

Figure 4: Plot of mQuine's test loss over 900 training epochs.

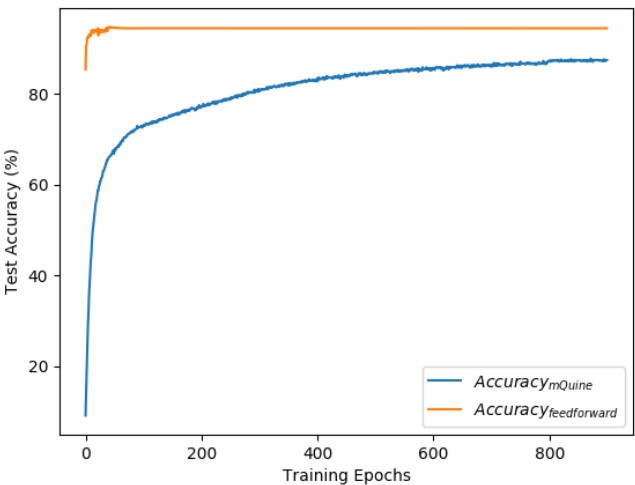

Figure 5: Plot of mQuine's test accuracy against a non-self-replicating baseline with the same architecture over 900 training epochs.

