# OpenReview forum: "Neural Network Quine"
_ICLR.cc/2018/Workshop — Reject_

### Official Review · AnonReviewer1 · 2018-03-02
**Quines are computer programs that print their own source code. The authors describe a way of building neural nets that print their own weights. Let's wait for rebuttal phase.**

**Rating:** 5
**Confidence:** 5

**Review:**

Quines are computer programs that print their own source code. The authors describe a way of building neural nets that print their own weights.

This is a very special case of what was done around 1992-1993: self-referential recurrent networks [1,2,3,4] not only read and write their own weights, but also can run entire computable learning algorithms or weight modification algorithms, to “learn how to learn”  (see also [7] for a more recent meta-learning application).

The authors write: "This overcomes the problem of Θ being larger than the output, since we are only outputting a scalar Θc for each coordinate c."

Likewise, the earlier work [1,2,3,4] used output units to encode weight addresses, to read and write them. There also is an outer-product-based way [5,6] of dramatically improving the ratio between addressable fast weights [0] and adaptive parameters.

Obviously the authors of this submission should point out what's new in their approach.

I'd like to see the revised version again.


REFERENCES


[7] S. Hochreiter, A. S. Younger, P. R. Conwell. Learning to learn using gradient descent. International Conference on Artificial Neural Networks. Springer, Berlin, Heidelberg, 2001.

[6] I. Schlag, J. Schmidhuber. Gated Fast Weights for On-The-Fly Neural Program Generation. Workshop on Meta-Learning, @NIPS 2017, Long Beach, CA, USA.

[5] J.  Schmidhuber. Reducing the ratio between learning complexity and number of time-varying variables in fully recurrent nets. In Proceedings of the International Conference on Artificial Neural Networks, Amsterdam, pages 460-463. Springer, 1993.

[4] J.  Schmidhuber. A neural network that embeds its own meta-levels. In Proc. of the International Conference on Neural Networks '93, San Francisco. IEEE, 1993.

[3] J.  Schmidhuber. An introspective network that can learn to run its own weight change algorithm. In Proc. of the Intl. Conf. on Artificial Neural Networks, Brighton, pages 191-195. IEE, 1993.

[2] J.  Schmidhuber. A self-referential weight matrix. In Proceedings of the International Conference on Artificial Neural Networks, Amsterdam, pages 446-451. Springer, 1993. PDF . HTML.

[1] J.  Schmidhuber. Steps towards `self-referential' learning. Technical Report CU-CS-627-92, Dept. of Comp. Sci., University of Colorado at Boulder, November 1992.

[0] J. Schmidhuber. Learning to control fast-weight memories: An alternative to recurrent nets. Neural Computation, 4(1):131-139, 1992.

---

### Official Review · AnonReviewer3 · 2018-03-09
**Interesting idea but no real results and not developed enough theoretically**

**Rating:** 2
**Confidence:** 5

**Review:**

The paper proposes to train neural networks to reproduce their own parameter vector.

While I find the motivations based on biological systems a little bit vague, the idea to train a neural network to reproduce its own weights is certainly original. Unfortunately it's not that clear to me why that would be a useful thing to do, neither does the paper present any results that support this.

The first experiment, in which the network is only trained for replication, seems somewhat ill-posed because a trivial solution to this task would be to learn a network in which all the weights are equal to zero. This leads me to think that the notion of self-replication should be connected to the information content of the weights in some way. The obtained error score on this task is hard to interpret without any baselines or variations of the task.

The second experiment shows that the performance on MNIST classification degrades when the replication criterion is added as an auxiliary task. This is quite obvious and it would have been an interesting finding if the opposite result was observed.

Pros: original idea
Cons: limited analysis, no insightful results

---

### Official Review · AnonReviewer2 · 2018-03-11
**nice idea**

**Rating:** 7
**Confidence:** 3

**Review:**

The paper is a nice workshop paper and I like the idea, also that it different from other papers I have seen in what it tries to achieve. I'm not sure self-replication would be a necessary capability or good first step to self-awareness or to self-improving but it's a possibility. Neural Turing machines also come to mind ...
The paper is a little sparse on technical details, and uses quite a lot of space (comparatively) for motivation and future work.

---

### Decision · Program_Chairs · 2018-03-20
**ICLR 2018 Workshop Acceptance Decision**

**Decision:**

Reject

**Comment:**

Based on the reviews, this paper has not been accepted for presentation at the ICLR workshop. However, the conversation and updates can continue to appear here on OpenReview.